

# Taurine stimulation of planarian motility: a role for the dopamine receptor pathway

Elisa J. Livengood[1], Robyn A. M. V. Fong[1], Angela M. Pratt[1],
Veronika O. Alinskas[1], Grace Van Gorder[1], Michael Mezzio[2],
Margaret E. Mulligan[2] and Evelyn B. Voura[3,4]

[1] Division of Environmental and Renewable Resources, State University of New York (SUNY) at Morrisville, Morrisville, New York, United States
[2] Department of Math and Science, Dominican University, Orangeburg, New York, United States
[3] Crouse Neuroscience Institute, Crouse Health at Crouse Hospital, Crouse Medical Practice, Syracuse, New York, United States
[4] Department of Neuroscience and Physiology, State University of New York Upstate Medical University, Syracuse, New York, United States

Corresponding author
Elisa J. Livengood,
livengej@morrisville.edu

## ABSTRACT

Taurine, a normal dietary component that is found in many tissues, is considered important for a number of physiological processes. It is thought to play a particular role in eye development and in the maturation of both the muscular and nervous systems, leading to its suggested use as a therapeutic for Alzheimer's and Parkinson's diseases. Taurine increases metabolism and has also been touted as a weight loss aid. Due to its possible benefits to health and development, taurine is added as a supplement to a wide array of products, including infant formula and energy drinks. Despite its pervasive use as a nutritional additive and implied physiological actions, there is little consensus on how taurine functions. This is likely because, mechanistically, taurine has been demonstrated to affect multiple metabolic pathways. Simple models and straightforward assay systems are required to make headway in understanding this complexity. We chose to begin this work using the planarian because these animals have basic, well-understood muscular and nervous systems and are the subjects of many well-tested assays examining how their physiology is influenced by exposure to various environmental, nutritional, and therapeutic agents. We used a simple behavioral assay, the planarian locomotor velocity test (pLmV), to gain insight into the stimulant properties of taurine. Using this assay, we observed that taurine is a mild stimulant that is not affected by sugars or subject to withdrawal. We also provide evidence that taurine makes use of the dopamine D1 receptor to mediate this stimulant effect. Given the pervasiveness of taurine in many commercial products, our findings using the planarian system provide needed insight into the stimulant properties of taurine that should be considered when adding it to the diet.

## INTRODUCTION

Taurine is an aminosulfonic acid (2-aminoethane sulfonic acid) that is readily available in foods such as meat, dairy, and fish but is also produced by the body (*Ripps & Shen, 2012*). Consequently, taurine is present in many tissues, particularly in skeletal muscle, as well as the heart, retina, and central nervous system, which are known to produce their own supply (*Ripps & Shen, 2012*; *De Luca, Pierno & Camerino, 2015*; *Bkaily et al., 2020*). In fact, taurine is reportedly the most abundant amino acid in the brain and retina (*Ripps & Shen, 2012*). As such, taurine is thought to be essential for neuronal, retinal, and muscular development (*Ripps & Shen, 2012*; *Li, Gao & Liu, 2017*; *Singh et al., 2023*). These ideas are reinforced by studies examining taurine deficiency, which was linked to maturational delays in cats and abnormalities in the visual cortex of primates (*Sturman et al., 1985*; *Neuringer et al., 1994*; *Palackal, Neuringer & Sturman, 1993*).

Due to these developmental influences, many have studied the benefit of taurine supplementation to support development, as well as for neuronal and muscular performance (*Almeida-Becerril et al., 2021*; *Ommati et al., 2024*; *Mohamed et al., 2023*; *Yu, Fan & Wu, 2024*). Since taurine concentrations decrease with age, it is thought that supplementation is required in adults who might not get sufficient amounts of the amino acid in their diet (*Verner, McGuire & Craig, 2007*; *Bkaily et al., 2020*; *Singh et al., 2023*). Taurine deficiency is considered by some to be a primary driver of aging as supplementation with taurine increases the health span of primates and the life span of worms and mice (*Kim, Do & Lee, 2010*; *Wu & Prentice, 2010*; *Marcinkiewicz & Kontny, 2014*; *Singh et al., 2023*). These findings lead to further research examining the role of taurine in the pathophysiology of neurodegenerative disorders, such as those by *Zhang et al. (2016)*, who reported reduced levels of taurine in the plasma of patients with Parkinson's disease, while *Basun et al. (1990)* documented decreases of taurine in cerebrospinal fluid of patients with Alzheimer's disease. This work aligns with other findings suggesting that taurine improves memory (*Kim et al., 2014*; *El Idrissi & L'Amoreaux, 2008*; *Tu et al., 2018*; *Bae et al., 2022*). Along these lines, taurine has been demonstrated to function as an antioxidant, mitigating oxidative stress (*Baliou et al., 2021*). Since Alzheimer's, Parkinson's, and Huntington's diseases share glutamate excitotoxicity, calcium imbalance, and oxidative stress, resulting in cell death, some have proposed that taurine could be helpful by functioning as a neuroprotectant, neuromodulator, and antioxidant in patients with these conditions (*Menzie et al., 2014*). Therefore, taurine dietary supplementation, either through energy drinks or nutritional additives, has been suggested for these cases (*Louzada et al., 2004*; *Oh et al., 2020*; *Kim et al., 2014*; *Menzie et al., 2014*).

Alongside the proposed therapeutic uses for taurine, there has been some indication that it may influence metabolism and function as a stimulant and, therefore, provide a means to support weight loss and improve cardiovascular function (*Jeukendrup & Randell, 2011*; *Rutherford, Spriet & Stellingwerff, 2010*; *Bkaily et al., 2020*; *Kim et al., 2019*; *Haidari et al., 2020*; *Cetin et al., 2023*). In contrast, earlier studies did not support these ideas, complicating our understanding of the role of taurine (*Baskin et al., 1974*;

*Whirley & Einat, 2008*). Regardless, the use of energy drinks and taurine supplementation as a stimulant for athletic performance continues to be a focus of study (*Rutherford, Spriet & Stellingwerff, 2010*; *Lim et al., 2018*; *Ozan et al., 2022*). As a result, there seem to be many factors at play, making it difficult to build a consensus on the usefulness of taurine for these purposes (*Chen et al., 2021*; *Kurtz et al., 2021*; *Buzdağlı et al., 2023*).

With so many disparate functions attributed to taurine, a considerable body of research has been focused on understanding how it exerts its effects. It has been suggested that taurine is a candidate neurotransmitter due to its association with membrane structures and influence on neuronal activity (*Wang, Xiao & Ye, 2005*; *Ripps & Shen, 2012*). Taurine is also known to cross the cell membrane cotransported with sodium (*Bkaily et al., 2020*). Some reports have linked taurine to multiple physiological pathways, including those involving the NMDA (N-methyl-D-aspartic acid), GABA (gamma-aminobutyric acid), glycine, glutamate, and dopamine receptor pathways (*Wu et al., 2009*; *Chan et al., 2014*; *El Idrissi & L'Amoreaux, 2008*; *Chan et al., 2013*; *Wang, Xiao & Ye, 2005*). Cellular uptake of taurine can influence dopamine receptor activation and increase levels of dopamine in the reward center (*Jiang et al., 2004*; *Chepkova, Sergeeva & Haas, 2005*; *Wang, Xiao & Ye, 2005*; *Ericson et al., 2006*; *Ericson et al., 2013*). The D1 receptor was particularly highlighted by *Suárez et al. (2014)* as being influenced by taurine. By examining D1 and D2 receptor agonists, *Chepkova, Sergeeva & Haas (2005)* provided evidence that taurine affects the dopamine reward pathway *via* corticostriatal synaptic transmission (*Bamford, Wightman & Sulzer, 2018*). These findings suggest that taurine plays a significant role in modulating dopamine-related processes.

With the purported broad physiological influence of taurine, nutritional supplementation with this amino acid is not unexpected but leads to other questions. Natural products such as taurine are classified as safe for consumption by governing organizations because they are derived from plants or occur naturally in the body (GRAS Notice 586, *U.S. Food and Drug Administration, 1998*; *Medina-Franco et al., 2012*). However, adding these substances to the diet as supplements in commercial formulations or in products such as energy drinks can introduce them to the body at higher concentrations or in combinations that would not normally be encountered, which may be detrimental to human health (*Ellermann et al., 2022*; *Kaur et al., 2022*). Furthermore, it is difficult to determine how these products function or even if they have an effect in these commercial combinations (*Alford, Cox & Wescott, 2001*; *Warburton, Bersellini & Sweeney, 2001*; *Giles et al., 2012*).

To gain insight into the complex physiological role of taurine, a straightforward animal model is necessary. Planarians provide one such useful paradigm. Planarians have long been used to understand development and regeneration (*Ivankovic et al., 2019*; *Reddien & Sánchez Alvarado, 2004*; *Reddien, 2018*). These studies have now matured to the point where investigators are exploring molecular and cellular mechanisms involved in regeneration and other complex physiological questions (*Reddien, 2018*). Furthermore, planarians have also historically been used to study the effects of environmental factors (*Voura et al., 2017*; *Hall, Morita & Best, 1986a*, *1986b*). These studies have featured drugs, a variety of toxins, as well as natural products (*Pagán, Coudron & Kaneria, 2009*;

*Moore, 1918*). Using simple animal models like the planarian in these cases allows for connections to be made between the basic biological pathways and the evolutionary potential of genes (*Voura et al., 2017*). Moreover, planarians undergo cephalization and share neurotransmitters and neuronal populations with the mammalian system, allowing for possible comparisons between species in neuropharmacological and neurotoxicology studies (*Cebrià et al., 2002*, *2007*; *Pagán et al., 2012*; *Hagstrom, Cochet-Escartin & Collins, 2016*). Planarians also exhibit a wide range of morphological and behavioral responses that can be measured and observed in a number of useful assays (*Raffa, Holland & Schulingkamp, 2001*; *Raffa, Stagliano & Tallarida, 2006*).

To follow up on our previous work, which examined the role of guarana as a stimulant, we used planarians and adapted the previously established planarian locomotor velocity assay (pLmV) to similarly test for taurine stimulation in these animals (*Moustakas et al., 2015*). Prior to our studies on guarana, the pLmV assay was demonstrated as an effective model for assessing the excitatory and withdrawal properties provided by cocaine and methamphetamines, as well as other products (*Raffa, Holland & Schulingkamp, 2001*; *Rawls et al., 2011*; *Gentile, Cebrià & Bartscherer, 2011*). Using the pLmV assay, we deduced that guarana had stimulant properties separate from those provided by caffeine (*Moustakas et al., 2015*). Since the stimulant properties of taurine are similarly unclear, we again used the pLmV assay to evaluate the excitatory properties of this substance and identify a possible mechanism behind these effects. Our findings suggest that taurine does function as a mild stimulant and provides evidence that the dopamine receptor pathway is connected to this response.

## MATERIALS AND METHODS

### Planarian husbandry

Planarians (*Dugesia* sp.) were purchased from Carolina Biological Supply Company (132950; Burlington, NC, USA). Refer to *Voura et al. (2020)* for a detailed description of planarian husbandry. Briefly, planarians were maintained in plastic food storage containers in spring water (Poland Spring). Planarians were fed twice weekly using frozen bloodworms or pieces of boiled eggs *ad libitum* for several hours. Afterward, the animals were placed into clean containers with fresh spring water or given a water change of spring water.

### Planarian locomotor velocity test

Planarians were acclimated to the laboratory environment and feeding schedule for a few weeks before being used for experimentation. The planarian locomotor velocity (pLmV) test, as established by R.B. Raffa and S.M. Rawls, was adapted for this study as we previously described (*Raffa, Holland & Schulingkamp, 2001*; *Raffa, Stagliano & Tallarida, 2006*; *Rawls et al., 2011*; *Ramoz et al., 2012*; *Moustakas et al., 2015*; *Voura et al., 2020*). Briefly, 1 week before a planned test, the animals were starved to restrict the influence on the pLmV assay to the substance being tested. Each worm was examined using a stereomicroscope prior to experimentation to ensure they were fully formed and without irregularities. The experimental size of the planarians ranged from 0.5 to ~1.0 cm.

Planarians were habituated in their respective test substance or combination at the appropriate assay concentration for either 2 min, 15 min, or 1 h. The worms were then transferred to the center of a 10 cm diameter Petri dish placed on grid paper, with lines spaced 0.5 cm apart, containing 20 mL Poland spring water or test solution. Planarian motility was monitored for 5 min with the exception of the initial stimulant study in which planarians were observed for 3 min, 5 min, and 1 h by assessing the number of grid lines crossed by the planarian during that time. Planarian motility was recorded by video camera for the duration of the experiment, and the recordings were reviewed to assess the grid counts (*Voura et al., 2020*). Test concentrations were deemed too high if the worms exhibited coiling or convulsive behavior during the habituation period. Tests were conducted at different times of the day to account for circadian behavioral variations between animals. Additionally, two or three different trained experimenters conducted each set of assays. No detectable change in water quality, such as pH, resulted from the substances studied at the concentrations examined. Each worm, including control specimens, was used only once.

## Stock solutions

For each test, stock solutions of taurine, sucrose, glucose, or dopamine receptor antagonists were diluted in spring water to the desired concentration unless otherwise noted. Taurine was purchased from Sigma-Aldrich (T0625; St. Louis, MI, USA) and diluted from a 3 mM stock made in spring water (Poland Spring). D-glucose or dextrose from Sigma Aldrich (D9434; St. Louis, MI, USA) was prepared as a 561 mM stock in spring water. Sucrose from Ward's Science (070302; Rochester, NY, USA) was prepared as a 20% stock solution in spring water. The D1 receptor antagonist, SCH 23390 hydrochloride or (R)-(+)-7-chloro8-hydroxy-3-methyl-1-phenyl-2,3,4,5-tetrahydro-1H-3-benzazepine hydrochloride, was from TOCRIS (0925; Bristol, United Kingdom) and diluted first as a primary 5 mM stock in spring water. This primary stock was diluted again to a secondary 0.5 mM stock in spring water. The D2 receptor antagonist, (S)-(−)-Sulpiride or S(−) sulpiride 5 (Aminosulfonyl)- N-[(1-ethyl-2pyrrolidinyl)methyl]-2- methoxy-benzamide was also purchased from TOCRIS (0895; Bristol, United Kingdom) and was dissolved in DMSO (470300-982; Ward's Science, Rochester, NY, USA) to make a primary 5 mM stock and then again as a 0.5 mM secondary stock solution. This secondary stock was diluted in spring water for experiments as needed so that both the D1 and D2 receptor antagonists had a final habituation concentration of 0.001 mM (see Inhibition Study below). Consequently, experiments using the D2 receptor antagonist resulted in DMSO being introduced to pre-habituation step at a concentration of 0.2%, requiring a 0.2% DMSO only experimental run and 0.2% DMSO followed by 0.03 mM taurine control/test (see Inhibition Study below).

## Stimulant study

Planarians were habituated in 5 mL of spring water or in 0.003, 0.01, 0.03, 0.1, 0.3, or 1 mM solutions of taurine by diluting the 3 mM stock solution as appropriate. The habituation lasted 2 min or 1 h as needed. The pLmV assay was then conducted for 3 and 5 min in

20 mL of the corresponding taurine concentration used for the habituation. All controls were run in spring water.

### Sugar study

Planarians were habituated in 5 mL of 0.1% sucrose alone, 0.1% sucrose with 0.03 mM taurine, 0.516 mM glucose with 0.03 mM taurine, or 5.16 mM glucose with 0.03 mM taurine for 2 min or 1 h by diluting the 20% sucrose, 561 mM D-glucose or 3 mM taurine stock solutions as required. The pLmV assay was then run for 5 min in 20 mL of the same solution used for habituation. All controls were run in spring water.

### Withdrawal study

Planarians were habituated in 5 mL of spring water or in 0.01 mM, 0.03 mM, and 0.1 mM solutions of taurine by diluting the 3 mM stock solution. The habituation times were 2, 5, or 15 min as required. The pLmV assay was then conducted for 5 min in 20 mL of spring water without taurine.

### Inhibition study

For inhibition studies using the D1 or D2 receptor antagonists, planarians were pre-habituated in 5 mL of spring water, 0.001 mM of the D1 receptor antagonist SCH 23390, or the 0.001 mM of the D2 receptor antagonist (S)-(−)-Sulpiride for 15 min prior to starting the pLmV assay. For these incubations, the 0.5 mM secondary stock solutions of the dopamine receptor antagonist were used (as described in Stock Solutions above). Since the solvent used for the D2 receptor antagonist stock solution was DMSO, an additional DMSO control (instead of water only) was necessary for these experiments. For this control, 10 μL of DMSO was diluted to 5 mL in spring water for the pre-habituation, which resulted in a 0.2% DMSO concentration. After the habituation in 0.2% DMSO, for the 2-min habituation the pLmV assay was conducted in spring water alone. For experiments assessing the effect of the D1 or D2 receptor antagonists alone, the 2-min habituation and pLmV assay were conducted in spring water. In each case, following the 15-min pre-habituation of the planarians in the required antagonist or control, planarians were dipped several times into a container with 2 mL of clean spring water to remove the pre-habituation solution before the 2-min pLmV assay habituation. For the 2-min habituation, planarians were placed in 5 mL of spring water for controls or 0.03 mM taurine for tests, which was used as needed. As with the other experiments, taurine was diluted using the 3 mM stock solution. The pLmV assay was then conducted for 5 min in spring water for controls or 0.03 mM taurine for the tests.

### Data analysis

Each experiment was conducted a minimum of three times, and each was typically done in triplicate, if not more, so that the final pLmV data was assessed using at least nine worms for each condition. Specific numbers of worms are referenced in the Results, in the figure legends, and a table (Table 1) referencing the total number of worms in each experiment is found in the link provided in the data availability section. All data points were normalized to their respective controls and expressed as the mean +/− the standard deviation. Each

**Table 1 Sample size by experimental condition.**

| Experimental assay | Control sample size | Test condition sample size | ANOVA degrees of freedom where applicable |
|---|---|---|---|
| Figure 1. pLmV by concentration—2 min | $n = 18$ | $n = 9$ for each concentration (0.003, 0.01, 0.03, 0.1, 0.3, 1.0 mM) | |
| Figure 1. pLmV by concentration—1 h | $n = 18$ | $n = 9$ for each concentration (0.003, 0.01, 0.03, 0.1, 0.3, 1.0 mM) | |
| Figure 2. Timecourse study by concentration—0.03 mM | $n = 10$ | $n = 15$ | df 5/69 |
| Figure 2. Timecourse study by concentration—0.1 mM | $n = 12$ | $n = 12$ | df 5/79 |
| Figure 2. Timecourse study by concentration 0.01 mM | $n = 8$ | $n = 12$ | df 5/62 |
| Figure 3A. Glucose—2 min | $n = 6$ | $n = 9$, $n = 9$ (0.561, 561 mM) | |
| Figure 3A. Glucose—1 h | n=6 | $n = 9$, $n = 9$ (0.561, 561 mM) | |
| Figure 3B. Sucrose alone—2 min | $n = 9$ | $n = 12$ | |
| Figure 3B. Sucrose and taurine—2 min | $n = 4$ | $n = 7$ | |
| Figure 3B. Sucrose alone—1 h | $n = 8$ | $n = 9$ | |
| Figure 3B. Sucrose and taurine—1 h | $n = 7$ | $n = 8$ | |
| Figure 4A. Withdrawal study—2 min | $n = 9$ | $n = 17$ | df 5/88 |
| Figure 4A. Withdrawal study—5 min | $n = 9$ | $n = 12$ | df 5/62 |
| Figure 4B. Withdrawal study—15 min | $n = 5$ | $n = 13$ | df 5/64 |
| Figure 5A. Dopamine receptor 1 inhibitor (D1) | $n = 10$ | $n = 12$ | |
| Figure 5A. Dopamine receptor 1 (D1) inhibitor with taurine | $n = 8$ | $n = 12$ | |
| Figure 5B. DMSO with 0.03 mM taurine *versus* the DSMO control | $n = 6$ | $n = 10$ | |
| Figure 5B. Dopamine receptor 2 inhibitor (D2) *versus* the DMSO control | $n = 8$ | $n = 12$ | |
| Figure 5B. Dopamine receptor 2 inhibitor (D2) with 0.03 mM taurine | $n = 7$ | $n = 12$ | |

data point was analyzed using an unpaired student's T-test or ANOVA, where appropriate, and was deemed significant if the corresponding $p$-value was less than 0.05. A *post hoc* analysis Tukey's test for ANOVAs was conducted on the stimulant, time course, and withdrawal assay results where appropriate. All statistical analyses were conducted using Microsoft Excel (Office Professional 2019).

## RESULTS

### Taurine stimulates planarian locomotor activity

The planarian locomotor velocity (pLmV) test was adapted to assess the stimulant properties of taurine. We began by testing a gauntlet of concentrations including 0.003, 0.01, 0.03, 0.10, 0.30, and 1.0 mM and observing planarian motility after incubation with taurine following a 2-min habituation (Fig. 1A). From the tested concentrations, the most effective stimulation occurred above 0.003 mM, with significant stimulation occurring at

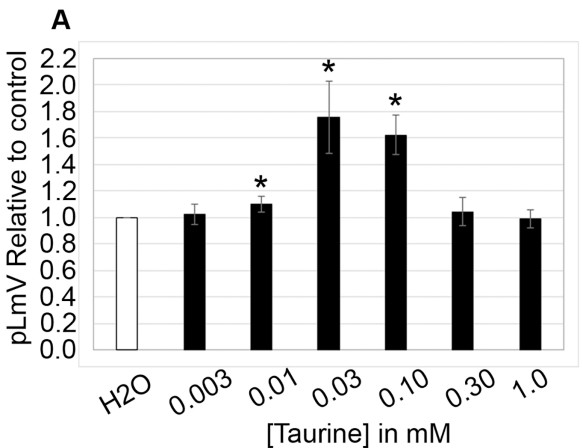

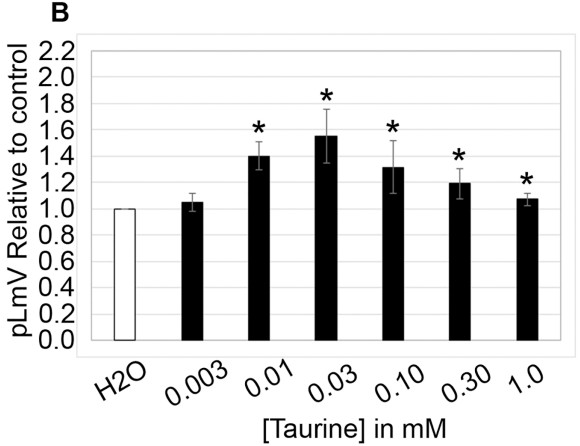

**Figure 1 pLmV by concentration gauntlet.** pLmV data at 3 min relative to the water control (A) following a 2-min or (B) 1-h habituation in taurine ($n = 9$ for each concentration and habituation time). In each case, the habituation and the pLmV assay were completed in the taurine concentration indicated. (A) An asterisk (*) indicates a significant difference from the water control ($\leq 0.05$) using an unpaired Student's T-test.

0.01, 0.03, and 0.10 mM concentrations (ANOVA $p < 0.001$; $n = 9$ for each concentration) when compared to the water control. To determine if a longer exposure to taurine would have added stimulation, we incubated planarians for 1 h in selected taurine concentrations (Fig. 1B). We observed a significant difference from the water control for concentrations of 0.01, 0.03, 0.10, 0.30, and 1.0 mM (ANOVA $p < 0.001$) after incubation for 1 h (Fig. 1B). Because the resultant stimulation was greater overall after 2 min, we opted for this incubation time for our studies. To decide how long to run our stimulation assays, a 5-min time course study was conducted to determine the duration of the stimulant effect (Fig. 2). For this, we focused on the three concentrations that offered the greatest stimulation as assessed by our initial experiments, which were 0.01, 0.03, and 0.10 mM. The concentrations of 0.01, 0.03, and 0.10 mM provided stimulation that was different from the water control ($p = 0.0099$; $<0.0001$; $0.0042$; $n = 12$; $n = 15$, $n = 13$, respectively). While the initial stimulation provided by the 0.03 mM concentration was higher overall relative to

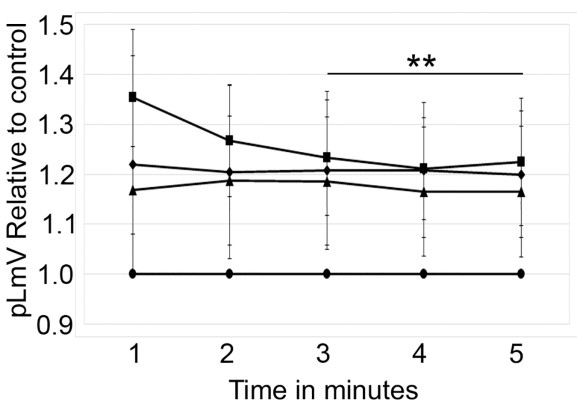

**Figure 2 Time course study by concentration.** pLmV time course data after a 2-min habituation in 0.01 mM taurine (triangles; $n = 12$), 0.03 mM taurine (squares; $n = 15$), and 0.1 mM taurine (diamonds; $n = 13$) with the respective concentrations maintained during the pLmV assay. The data are relative to the water control (circles). ANOVA results indicate that the lines for each concentration are statistically different from the control ($p \leq 0.05$; not indicated on the graph). The double asterisk (**) indicates these time points are significantly different ($\leq 0.05$) using an unpaired Student's T-test from the 1-min time point for the 0.03 mM time course data.

the 0.01 and 0.10 mM concentrations, the effect of all three concentrations stabilized by 3 min, indicating that this time was satisfactory to run our additional pLmV assays.

## Sugars do not influence the effect of taurine on pLmV

To determine if sugars affected taurine stimulation, we tested whether sucrose and glucose affected the pLmV results with 0.03 mM taurine. The glucose concentrations used were identical to those used in our previous guarana study and were based on the amount of glucose found in typical energy drink formulations (*Moustakas et al., 2015*). Planarians were exposed to 0.561 mM glucose with 0.03 mM taurine or 5.61 mM glucose with 0.03 mM taurine, and the pLmV was assessed at 3 min (Fig. 3A). Motility was significantly greater when compared to the water control for taurine with glucose at 0.561 mM at both 2 min and 1 h ($p = 0.0011$, $\leq 0.001$, respectively, $n = 9$ for each time). At the higher concentration, 5.61 mM with 0.03 mM taurine motility was only significantly different from the water control after the 1-h habituation, but not after 2 min. When comparing the results for the different concentrations of glucose after the same habituation times, we observed that these were also significantly different with $p < 0.001$, for 2 min and $p = 0.012$, for 1 h, respectively. However, compared to the stimulation with both glucose and taurine, glucose did not provide additional stimulation after a 2-min habituation over taurine alone (Fig. 3A). Furthermore, when examining the data after the 1-h habituation, the addition of glucose was inhibitory when compared to the results with taurine alone (Fig. 3B). Examining the effect on motility with sucrose alone or with sucrose and 0.03 mM taurine, we did not observe any significant difference from the water controls (Fig. 3B). For the sucrose assays at 2 min and 1 h for sucrose alone $n = 12$ and $n = 9$, respectively, and for the assays with sucrose and taurine $n = 7$ at 2 min and $n = 8$ at 1 h.

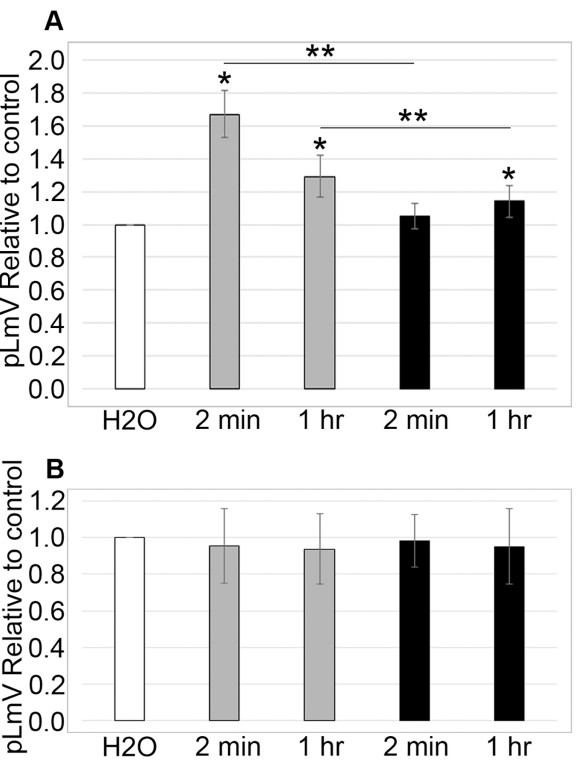

**Figure 3 Sugar study.** pLmV data at 3 min following (A) a 2-min or 1-h habituation in 0.561 mM glucose with 0.03 mM taurine (grey bars; $n = 9$ for both times), or 5.61 mM glucose with 0.03 mM taurine (black bars; $n = 9$ for both times) and (B) a 2-min or 1-h habituation in 0.1% sucrose alone (grey bars; $n = 12$ for the 2-min data and $n = 9$ for the 1-h data), or 0.1% sucrose with 0.03 mM taurine (black bars; $n = 7$ for the 2-min data and $n = 8$ for the 1-h data). The pLmV assay was run in the presence of 0.03 mM taurine for each case. Symbols indicate significance at $p \leq 0.05$ where the an asterisk (*) indicates a difference from the water control, and a double asterisk (**) indicate a difference between the different concentrations of glucose after the same habituation time (between the concentrations after 2 min or 1 h), using an unpaired Student's T-test.

## Taurine stimulation is short-lasting and not associated with withdrawal

To determine if stimulation was associated with a withdrawal effect, planarians were habituated in 0.03 mM taurine for 2 or 5 min prior to an assessment of pLmV run in spring water alone for 5 min ($n = 17$ and $n = 12$) (Fig. 4A). We did not observe any detectable withdrawal effect (as would be exemplified by pLmV results below that of the spring water control) and determined that the stimulation was lost prior to the 1-min point once the worms were removed from taurine. By comparison, when planarians were habituated for 15 min in 0.03 mM taurine (Fig. 4B) with pLmV in spring water (ANOVA; $p = 0.00396$) we observed a better-sustained stimulation compared to habituation in spring water alone for the full 5 min ($n = 13$ for each and $p = 0.0052, 0.0063, 0.0052, 0.0007, 0.00065$; for the 1, 2, 3, 4 and 5-min points, respectively). From this, we determined that there was a gradual drop off of the stimulation, causing the data to be significantly different between 1 and 4 min ($p = 0.048$). However, we did not observe a withdrawal effect following the 15-min habituation.

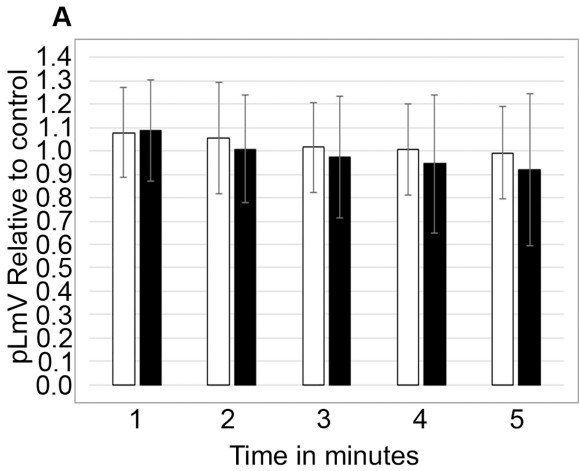

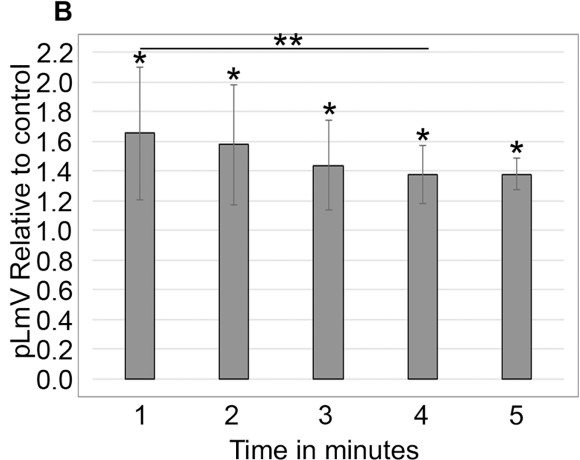

**Figure 4 Withdrawal study.** 5-min pLmV time-course data (A) after a 2-min (white bars; $n = 17$) or 5-min (black bars; $n = 12$) habituation in 0.03 mM taurine, or (B) after a 15-min habituation in 0.03 mM taurine (grey bars; $n = 13$). For these data the pLmV assay was completed in water alone. Symbols indicate significance at $p \leq 0.05$, where the an asterisk (*) indicates a difference from the water control, and a double asterisk (**) indicates a difference between the 1 and 4-min time points using an unpaired Student's T-test.

## Dopamine D1 receptor is associated with taurine stimulation

Using antagonists targeting the D1 and D2 receptors, we evaluated whether the dopamine receptor pathway contributed to the taurine-based stimulation of planarian motility. We observed significant inhibition using the D1 receptor inhibitor SCH 23390 ($p = 0.02$; $n = 12$), which was partially recovered with the addition of 0.03 mM taurine ($n = 12$; Fig. 5A). We did not detect any inhibition of planarian motility using the D2 receptor antagonist (S)-(−)-Sulpiride ($n = 12$ without taurine, $n = 12$ with taurine). However, inhibition using the D2 receptor inhibitor was complicated by the necessity of using DMSO as the solvent. Therefore, we also ran the experiment with DMSO alone and compared the D2 receptor inhibition to the DMSO control, which also affected the pLmV assay results in that we did not observe a taurine stimulant effect when DMSO alone was used in the pre-habituation step (Fig. 5B).

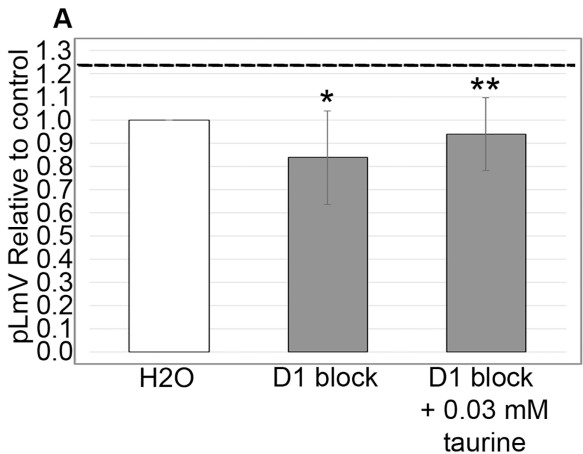

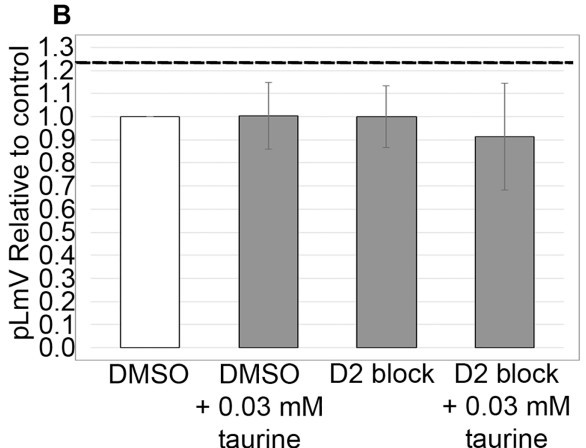

**Figure 5 Dopamine receptor inhibitor study.** pLmV data following a 15-min pre-incubation with (A) 0.001 mM of D1 receptor inhibitor, or (B) 0.001 mM of a D2 receptor inhibitor relative to the water only control for A or a DMSO control for B (white bars). The dotted line represents the pLmV data relative to a water control without an inhibitor at the 3-min mark of a pLmV assay with 0.03 mM taurine following a 2-min habituation with 0.03 mM taurine ($n = 15$). The data are shown after a 15-min pre-incubation with the D1 antagonist alone following a 2-min habituation in water (D1 block; $n = 12$) or in 0.03 mM taurine (D1 block + 0.03 mM taurine; $n = 12$) with the grey bars in (A) and with the D2 antagonist alone following a 2-min habituation in water (D2 block; $n = 12$) or in 0.03 mM taurine (D2 block + 0.03 mM taurine; $n = 12$) with the grey bars in (B). Also shown in (B) is a gray bar depicting the pLmV relative to the DMSO control following a 15-min pre-incubation in DMSO alone (the vehicle for the D2 receptor inhibitor) followed by a 2-min habituation in 0.03 mM taurine (DMSO + 0.03 mM taurine; $n = 10$). These data were selected from a 5-min time-course study run for each condition. Symbols indicate significance at $p \leq 0.05$ where an asterisk (*) is compared to the water control, and a double asterisk (**) is compared to the data including taurine as represented by the dotted line for an unpaired Student's T-test.

## DISCUSSION

Planarians have been used as model organisms in studies focused on understanding development, regeneration, toxicology, and pharmacology because of their complex nervous systems, which exhibit diverse behavioral responses comparable to those of vertebrates (*Pagán, 2017*; *Gentile, Cebrià & Bartscherer, 2011*). Specifically, the planarian nervous system uses neurotransmitters such as GABA, glutamate, dopamine, serotonin,

and acetylcholine, making them suitable for studies focused on the varied effects of neuromuscular biomodulators (*Nishimura et al., 2008*; *Voura et al., 2020*). Planarian motility assays, particularly the popular planarian locomotor velocity test (pLmV), have been used for studies examining stimulation, addiction, and withdrawal (*Raffa, Holland & Schulingkamp, 2001*; *Rawls et al., 2011*).

We also made use of the pLmV assay in previous work examining the stimulant properties of guarana (*Moustakas et al., 2015*). Importantly, for this and other behavioral studies involving planarians, both the experimental environment and planarian diet can be strictly controlled, limiting the exposure of the animals to the factor under investigation. Furthermore, because planarians can be starved for a limited amount of time without affecting their basal metabolic rate, it is possible to ensure the results are not subject to confounding dietary factors (*Felix et al., 2019*). In this case, starving the planarians before exposure to taurine ensured that taurine was the primary variable influencing motility in our studies. With so many functions and pathways at work relating to taurine physiology, taking advantage of these added controlling features made the planarian model particularly useful in our work. Since taurine has been suggested as a possible therapy for neuromuscular disorders, understanding how exposure to the amino acid influences movement with the pLmV assay is useful—even more so since planarians have been proposed as a model to study the influence of therapies for Parkinson's disease (*Prokai et al., 2013*). Given that taurine and guarana are included in combination with other natural products in food and beverages, such as energy drinks, it is important to investigate the biological activities of these substances in more detail due to their potential additive effects (*García et al., 2017*). Notably, as we demonstrated with our studies on guarana, planarians can significantly help our understanding of these effects (*Moustakas et al., 2015*).

Like guarana, taurine is a common additive in many energy drink formulations (*Costantino et al., 2023*). Taurine, however, is known to affect multiple biological systems and is found ubiquitously in the body, so the influence of taurine can be widespread (*Wu et al., 2009*; *Chan et al., 2014*; *El Idrissi & L'Amoreaux, 2008*; *Chan et al., 2013*). Understanding the stimulant effect of taurine is important because it is suggested as a possible therapeutic for neurological and neurodegenerative disorders—possibly using energy drinks (*Oh et al., 2020*; *Jangra et al., 2024*) and as a supplement for muscle recovery (*Kurtz et al., 2021*; *Ommati et al., 2019*). Because taurine has been suggested as a weight loss aid, we decided to begin our investigation by first determining if taurine functions as a stimulant in the planarian (*Jeukendrup & Randell, 2011*; *Rutherford, Spriet & Stellingwerff, 2010*; *Bkaily et al., 2020*; *Kim et al., 2019*; *Haidari et al., 2020*; *Cetin et al., 2023*). Indeed, some reports do suggest that taurine might have stimulant properties (*Chen et al., 2021*; *Kurtz et al., 2021*; *Buzdağlı et al., 2023*; *Wang, Xiao & Ye, 2005*). We began by testing taurine concentrations that were similar to those used for our studies of guarana as a starting point (Fig. 1). After testing a gauntlet of concentrations, we did observe that taurine produced a mild stimulant effect (Fig. 2), with the most significant stimulation occurring at concentrations of 0.01, 0.03, and 0.10 mM (Fig. 1). When comparing our findings with those of vertebrate models like mice, there has been conflicting evidence of

taurine acting as a stimulant. In one study, they found no evidence of taurine acting as a stimulant or having any effect on anxiety and depression-like behaviors (*Whirley & Einat, 2008*). However, when considered in combination with caffeine, taurine was observed to improve the cognitive function and motor ability of sleep-deprived mice (*Du et al., 2022*). Other work with taurine has focused on the recovery of skeletal muscle and the reduction of oxidative stress post-exercise in rats (*Silva et al., 2011*). Despite having no known specific receptor, taurine reportedly affects several signaling pathways (*Ripps & Shen, 2012*). To approach the question of how taurine could elicit the observed stimulant effect, we decided to begin first by examining the dopamine receptor pathway. We chose this pathway because of prior evidence that planarians respond to dopaminergic antagonists with changes in motility, which is amenable to observation using the pLmV assay (*Raffa, Holland & Schulingkamp, 2001*). There is also demonstrated use of inhibitors for the D1 and D2 receptor pathways in the planarian model (*Mohammed Jawad, Hutchinson & Prados, 2018; Tashiro et al., 2014; Zhang et al., 2013*). We observed a notable inhibition of planarian motility using the D1 receptor antagonist (SCH 23390), which was partially recovered in the presence of taurine. The inhibitor likely had a higher affinity for the D1 receptor than taurine, as evidenced by the partial recovery of motility with taurine and the antagonist when compared to that using the SCH 23390 alone (Fig. 5A). Additionally, we utilized the pLmV assay to assess the effect of the D2 receptor antagonist ((S)-(−)-Sulpiride). Others used higher concentrations of this drug and observed planarian immobility, corkscrew, and convulsive behaviors, which are not amenable to our assay system because it requires the forward movement of the animals (*Buttarelli et al., 2000*). As a result, we used a lower concentration than reported by *Zhang et al. (2013)*, which did not result in an inhibition of planarian motility (Fig. 5B). In contrast, *Raffa, Holland & Schulingkamp (2001)* observed some inhibition using concentrations of this antagonist that were lower than those used by us.

In addition to stimulants like guarana, taurine, and caffeine, energy drinks are typically also high in sugar, containing glucose, sucrose, and high fructose corn syrup (*Al-Shaar et al., 2017*). Following our prior work examining the combined effect of glucose with both caffeine and guarana, we followed a similar methodology to determine the impact of glucose and sucrose on our observed mild stimulant effect provided by taurine using the pLmV assay. The concentrations we chose to assess the effect of glucose were based on our prior publication, wherein we investigated concentrations ranging from 0.561 to 561 mM. These concentrations were based on the average reported glucose concentration in various energy drink formulations (*Moustakas et al., 2015*). In our previous study, we focused on the low and high-end concentrations (0.561 and 5.61 mM) after 2 min and 1 h of exposure. As a result, we applied these standards to our current study using taurine. We determined that glucose at the lower concentration of 0.561 mM in combination with taurine did not support a significant increase in planarian locomotor velocity beyond that of taurine stimulation alone (Figs. 1 and 3A). Similarly, at the higher 5.61 mM concentration, glucose in combination with taurine did not provide additional stimulation supporting motility (Fig. 3A). Instead, this higher concentration proved inhibitory to planarian movement but was not associated with any observable convulsive or other behaviors that might indicate

some impediment to planarian forward progress. Unlike what we observed for glucose and guarana, these findings agree with our prior work that glucose alone and glucose with caffeine do not promote planarian locomotor velocity (*Moustakas et al., 2015*). For sucrose, we based our work on concentrations established by others, which made use of 1% and 0.1% solutions in studies using planarians (*Ouyang et al., 2017*; *Zhang et al., 2013*). In our hands, 1% sucrose was not well-tolerated, resulting in convulsions and planarian death. *Ouyang et al. (2017)* similarly observed that high concentrations of sucrose of 10% produced a stereotypical response (C-shapes) and reduced motility. Other work with sucrose and planarians documented a conditioned place preference response to sucrose concentrations as high as 10%, which incidentally involved the dopamine reward pathway (*Mohammed Jawad, Hutchinson & Prados, 2018*). We determined that 0.1% sucrose was tolerated but did not observe a significant increase in motility over that of the water control (Fig. 3B). Sucrose also did not have any additive effect with taurine on planarian motility. We used the 0.03 mM concentration of taurine for all experiments examining the sugar concentrations, which provided the most stimulation in our initial experiments (Fig. 1).

Dopamine is associated with movement regulation and also has a role in the reward pathway (*Juárez Olguín et al., 2016*). Since dopamine agonists often produce a withdrawal effect, we also examined whether our observed taurine stimulation resulted in a noticeable withdrawal (*Yu & Fernandez, 2017*). The use of the pLmV assay is also well-established in withdrawal studies, with planarians often showing a reduction in locomotor activity below that of the water control following the removal of a stimulant—considered the hallmark withdrawal response (*Ramoz et al., 2012*; *Sacavage et al., 2008*). At the shorter habituation periods of 2 and 5 min in taurine, there was no detectable withdrawal effect, with stimulation being lost quickly but not dipping below that of the water control (Fig. 4A). In contrast, when planarians were habituated for 15 min in taurine, there was evidence of sustained stimulation but no associated withdrawal (Fig. 4B). From these experiments, we determined that the stimulation provided by taurine is short-acting but can be extended by longer exposure times without an apparent withdrawal effect, which is of interest when considering using taurine as a therapeutic.

The limitations of our study are related both to the chosen model, the assay, and the receptor. While the planarian remains an excellent invertebrate model for exploring the effects of natural substances such as stimulation and withdrawal because the exposure to the additives being tested can be controlled, taurine, unlike other natural products, may exist endogenously within the animal (*Ripps & Shen, 2012*). However, this possibility does not detract from our findings that adding supplemental taurine does stimulate planarian motility. Additionally, we relied on available inhibitors to explore the dopamine receptor connection with taurine, and the available D2 inhibitor (S(−) sulpiride) required the use of DMSO as a solvent. With the addition of DMSO, the stimulant effect was no longer observed (Fig. 5B). The alternative solvent, ethanol, would also have caused difficulty with the chosen assay since ethanol can be used to immobilize planarians and could have provided a similar detrimental effect and/or caused us to question our findings using this inhibitor (*Stevenson & Beane, 2010*).

One limitation of this type of assay is that using the planarian locomotor velocity assay requires data collectors to manually count the gridlines the planarians crossed instead of using a machine-learning vision system. Proper training of those collecting data is critical, which is why we formalized that training process as described in detail previously (*Voura et al., 2020*). Additionally, the use of multiple individuals collecting data and doing counts blinded following the assay using the recorded video allowed for some mitigation of possible bias. In this way, the planarian locomotor velocity assay remains an accessible method of determining planarian movement as a function of stimulation or withdrawal (*Raffa, Holland & Schulingkamp, 2001*; *Raffa, Stagliano & Tallarida, 2006*).

Finally, taurine stimulation can also be mediated through other pathways not explored in this work. Reports by others using different animal models provide evidence that taurine may use multiple physiology pathways, such as those involving NMDA, GABA, glycine, and glutamate, in addition to the dopamine receptor pathways (*Wu et al., 2009*; *Chan et al., 2014*; *El Idrissi & L'Amoreaux, 2008*; *Chan et al., 2013*, *Louzada et al., 2004*). Taurine also influences calcium homeostasis and has been shown to affect calcium channel activation (*Li et al., 2012*; *Schaffer, Solodushko & Kakhniashvili, 2002*; *Wu & Prentice, 2010*). It has also been reported that taurine can pass through the cell membrane *via* the sodium/chloride-dependent taurine transporter (TauT) (*Liu et al., 1992*). These findings highlight the many ways in which taurine can move into cells and influence multiple receptor pathways. Therefore, it is conceivable that taurine-mediated stimulation of planarian motility involves more than the dopamine D1 receptor pathway. To that end, planarians would provide a good model organism for exploring the influence of taurine on calcium homeostasis and channel activation. Additionally, when it comes to understanding the neuroprotective and regenerative effects of taurine, the planarian model could also be particularly useful due to the well-known regenerative capabilities of this organism (*Ivankovic et al., 2019*).

## CONCLUSIONS

Using the planarian model, we determined that taurine is a mild, short-acting stimulant that takes advantage of the dopamine D1 receptor pathway without a noticeable withdrawal effect. Many studies focused on the use of taurine in combination with ethanol and caffeine, however, the power of the planarian model is that we can truly examine the effects of taurine alone (*Ginsburg & Lamb, 2008*). This helps to limit the confounding effects of other substances and helps determine the mechanism of action. This study serves as a foundation for exploring those effects further. Understanding how taurine might function as an additive and stimulant is important with its continued use as a supplement and in energy drinks. Given the proposed therapeutic use of taurine for weight loss, improving memory, aiding athletic performance, and supporting patients with neurodegenerative and muscular disorders, understanding the stimulant properties provide necessary insight into the many described functions of taurine.

# ACKNOWLEDGEMENTS

We would like to thank Dimitros Moustakas for suggesting the study of taurine as a possible stimulant, and Nicole M. Shantel for contributing to the 15-min taurine withdrawal studies. We would also like to thank and acknowledge the numerous undergraduate students who participated in either planarian husbandry and care or norming of methods, such as Rowan Coates, Kellis Morgan, and Rachel Soong.

### Funding
The authors received no funding for this work.

### Competing Interests
The authors declare that they have no competing interests.

### Author Contributions
- Elisa J. Livengood performed the experiments, analyzed the data, prepared figures and/or tables, authored or reviewed drafts of the article, and approved the final draft.
- Robyn A. M. V. Fong performed the experiments, prepared figures and/or tables, authored or reviewed drafts of the article, and approved the final draft.
- Angela M. Pratt performed the experiments, prepared figures and/or tables, and approved the final draft.
- Veronika O. Alinskas performed the experiments, prepared figures and/or tables, and approved the final draft.
- Grace Van Gorder performed the experiments, prepared figures and/or tables, and approved the final draft.
- Michael Mezzio performed the experiments, prepared figures and/or tables, and approved the final draft.
- Margaret E. Mulligan analyzed the data, prepared figures and/or tables, and approved the final draft.
- Evelyn B. Voura conceived and designed the experiments, analyzed the data, prepared figures and/or tables, authored or reviewed drafts of the article, and approved the final draft.

### Data Availability
The data is available at Harvard Dataverse: Voura, Evelyn, 2024, "Planarian Locomotor Velocity and Taurine", https://doi.org/10.7910/DVN/NOJICB, Harvard Dataverse, V1.

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
