# Peer review of "Taurine stimulation of planarian motility: a role for the dopamine receptor pathway"

_PeerJ, doi:10.7717/peerj.18671_

## Round 0.1 · original submission · Major Revisions

Dear Authors:

Reviewers suggest major reviews to improve the study on taurine-induced planarian motility. Key points include clarifying the use of planarians as models, specifying experiment duration, and ensuring consistent DMSO concentrations. The rationale for the methods is considered underdeveloped, and comparisons with previous vertebrate studies would strengthen the argument. Additionally, there are concerns about statistical reporting, including degrees of freedom and post-hoc tests. The reviewer also suggests the need to address inter-observer reliability as a limitation in future studies. So, the decision is Major Reviews.

Regards,

Dr. Manuel Jiménez

·

Basic reporting

RE: Taurine stimulation of planarian motility: a role for the dopamine receptor pathway (#104692)

This is a quite interesting study that could contribute to the further expansion of the literature regarding planarians as an animal model in pharmacology and toxicology. However, there are issues with the submission that must be addressed before it is accepted for publication. The main issues are regarding the experimental design and clarity of reporting, as outlined below. I am confident that the authors will address these issues to move forward with the publication of the work.

1. BASIC REPORTING

Lines 290-292: The authors state that “Planarians have been used as model organisms in studies focused on understanding development, regeneration, toxicology, and pharmacology because of their complex nervous systems, which exhibit diverse behavioral responses comparable to those of vertebrates…”.

Rev comment: This is partially correct. The relative complexity of the planarian nervous system is the logical justification for their use in pharmacology, toxicology and neuroscience. However, planarians were extensively used in developmental and regeneration biology long before it was even recognized that these organisms had a bona fide brain. The authors should revise this statement and add relevant references to support their claim.

Experimental design

2. EXPERIMENTAL DESIGN and RESULTS
Line 151: The authors state that “Planarian motility was monitored for the required timing of the assay….”.

Rev comment: What was the duration of each experiment? Perhaps I missed it and if so, please accept my apologies, but this needs to be explicitly reported as time factors are pharmacologically relevant. As currently written, the habituation times are clearly reported, but not the actual observation time of the motility experiments. I want to say that each assay is 5 minutes, but I am not certain. The authors should clarify this point.

Lines 173-174 and Line 404: The authors state that the sulpiride analogues were dissolved in DMSO in a “…primary 5 mM stock and then again as a 0.5 mM secondary stock solution.”

Rev comment: Even though the authors report that they tested worms with 0.2 % DMSO (10 uL in 5 mL of water), it is not clear whether the concentration of DMSO is constant across experiments as reported. Furthermore, in line 404: the authors state “…we determined that DMSO decreased planarian motility (Fig. 5B).” As shown, this figure does not show any apparent DMSO-induced motility decrease as the bars are normalized to its control, including the control bar, which seems to be normalized to itself. Also, 0.2% DMSO should not significantly decrease planarian motility, unless the worms are extremely small (~0.5cm or smaller). It may be useful for the authors to report the average planarian size in this study. DMSO effects in planarians have been reported. There are published papers that quantitatively explored the DMSO effects on planarian motility (an especially informative one is Stevens et al (2015)), and the information in said papers might be useful to the authors.

Line 372: The authors state that “…1% sucrose was not well-tolerated, resulting in convulsions and planarian death.”

Rev comment: How was planarian death assessed?

Figure 2 legend: The authors stated that “…ANOVA results indicate that the lines for each concentration are statistically different from the control…”.

Rev comment: The symbols are connected with no linear (or otherwise) regression analysis reported. As such, it is unclear to me why do they allude to “lines” amenable to ANOVA analysis. Perhaps the authors refer to data sets? This point must be clarified.

Validity of the findings

3. VALIDITY OF THE FINDINGS
As mentioned above, I find this to be an interesting study, and I look forward to seeing the revised manuscript.

Additional comments

4. General comments
Thank you for the opportunity of evaluating this paper.

Reviewer 2 ·

Basic reporting

The authors describe several experiments designed to assess whether taurine affects motility in planarians. In addition to testing the dose-dependent effects of taurine, they examined the possibility of interaction with sucrose and glucose, evaluated time course of drug action, and looked at the possible modulation of taurine’s effects through dopaminergic activity by testing in combination with dopamine inhibitors. Overall, I have a favorable assessment of the work. It is mostly well written, the methodology seems sound, and the data have the potential to make a contribution to the literature.

However, I would recommend requesting a revision of this manuscript. The introduction provides a comprehensive overview of the physiological effects of taurine, potential medical uses, and possible neurological mechanisms of action. It is far reaching. But, the rationale for the methods of the current study seems a bit rushed and truncated in the last paragraph. In my opinion, the rationale for studying the effect of taurine on motility in planarians deserves further development. Sell the reader on the notion that the planarian model can provide understanding beyond what has been learned with basic research on the relationship between taurine and locomotor activity in vertebrates. Along those lines, comparing and contrasting current results with those obtained from previous investigations of taurine on vertebrate motor behavior would be a nice addition to the discussion. Another thing to consider in terms of the rationale is that the introduction goes into taurine’s relationship with nutritional deficits, dementia, and athletic performance. However, the conclusion seems to focus mostly on energy drinks and possible interactions of taurine with other stimulants. However, such possible interactions, although potentially important, were not tested in the current investigation. Making clinical recommendations of administering taurine as a standalone supplement is, in my opinion, going quite a bit beyond the current data.

Experimental design

In the method, I think it is appropriate to include the number of planarians tested for every condition instead of stating it was at least 9. This could be accomplished in a table

Validity of the findings

I believe the reporting of the statistical analyses are incomplete. For example, it is appropriate to include the degrees of freedom for an ANOVA. Did the authors run post-hoc analyses to compare motility among doses in addition to taurine vs control?

In general, I think the figures are well done. Unless I am misunderstanding something, it seems redundant to include a bar for the control data in the figures given that the results are expressed as a proportion of control (the bars for H20 will always equal 1 and therefor it is not necessary to graph those data)

Additional comments

This investigation has a limitation common in most of the work I have read regarding planarian motility. Human observers were used to record the data, yet no effort was made to assess inter-observer reliability. Therefore, the possibility of bias is very real Although I see this as a considerable shortcoming, I do not wish to hold the authors to a higher standard than that which is commonly published. However, I highly recommend that the authors do explicitly state that the lack of a reliability check with independent observers is a limitation that should be remedied in future investigations.

---

## Round 0.2 · Minor Revisions

Dear Co-Authors:

Thank you for submitting your manuscript titled: "Taurine stimulation of planarian motility: a role for the dopamine receptor pathway". After review, we consider "minor reviews" - please address these remaining issues.

Best regards

Dr. Manuel Jiménez

·

Basic reporting

I am satisfied with the modifications in the manuscript. However, I would like to suggest the correct spelling of some of the references. “Riddien” should be “Reddien”. “Riddien and Sanchez” should be “Reddien and Sánchez Alvarado”. “Pagan” Should be “Pagán”. These are just the ones that jumped at me. I encourage the authors to revise their references in deference to the authors whose work they cited.

Experimental design

Satisfied

Validity of the findings

Satisfied

Additional comments

Satisfied

Reviewer 2 ·

Basic reporting

The authors effectively responded to my primary concerns with the previous manuscript. In the current version, they have included arguments justifying the use of planarians as an animal model and briefly compared their results with those obtained with vertebrates in prior research. In addition, they omitted the clinical recommendations which I believed suffered from a bit too much extrapolation.

Experimental design

The authors included a table as I suggested in my previous review.

Validity of the findings

The authors have included some further details regarding their statistical analysis. They should still report what type of post-hoc tests were used. Because the data are reported as proportion of control, I still think it is redundant to include the control data, which must have a value of 1, on the graphs in Figures 1-3.

Additional comments

As I suggested, the authors have addressed possible limitations using human observers.

---

## Round 0.3 · accepted · Accept

Dear Authors,

We are pleased to inform you that your manuscript titled “Taurine stimulation of planarian motility: A role for the dopamine receptor pathway” has been accepted for publication in PeerJ. The reviewers and editorial team found your study to be highly insightful, contributing significantly to the understanding of taurine’s role in neurophysiology and its potential applications.

We commend your efforts and the quality of your research and look forward to publishing your work. Congratulations on this accomplishment!

Sincerely,

Dr. Manuel Jiménez